# Adequacy and Sources of Protein Intake among Pregnant Women in the United States, NHANES 2003–2012

**DOI:** 10.3390/nu13030795

**Published:** 2021-02-28

**Authors:** Mary M. Murphy, Kelly A. Higgins, Xiaoyu Bi, Leila M. Barraj

**Affiliations:** Exponent, Inc., Center for Chemical Regulation and Food Safety, Washington, DC 20036, USA; khiggins@exponent.com (K.A.H.); xbi@exponent.com (X.B.); lbarraj@exponent.com (L.M.B.)

**Keywords:** dietary intake, nutrient intake adequacy, macronutrient, protein, maternal diet, pregnancy

## Abstract

Limited information is available on protein intake and adequacy of protein intake among pregnant women. Using data from a sample of 528 pregnant women in the National Health and Nutrition Examination Surveys (NHANES) 2003–2012, usual intake of protein (g/day and g/kg body weight (bw)/day) and prevalence of intake below the Estimated Average Requirement (EAR) by trimester of pregnancy were calculated using the National Cancer Institute method. Percent contributions to protein intake by source (i.e., plant and animal, including type of animal source) were also calculated. Mean usual intake of protein was 88 ± 4.3, 82 ± 3.1, and 82 ± 2.9 g/day among women in trimester 1, 2, and 3 of pregnancy, respectively, or 1.30 ± 0.10, 1.35 ± 0.06, and 1.35 ± 0.05 g/kg bw/day, respectively. An estimated 4.5% of women in the first trimester of pregnancy consumed less protein than the EAR of 0.66 g/kg bw/day; among women in the second and third trimesters of pregnancy, 12.1% and 12.8% of women, respectively, consumed less protein than the EAR of 0.88 g/kg bw/day. Animal sources of protein accounted for approximately 66% of total protein. Findings from this study show that one in eight women in the second and third trimesters of pregnancy have inadequate intake of protein. Pregnant women should be encouraged to consume sufficient levels of protein from a variety of sources.

## 1. Introduction

The importance of nutrition during pregnancy for both maternal and child health outcomes is well recognized [1]. Nutrient needs increase during pregnancy to support fetal growth and development along with alterations in maternal metabolism and tissue development [2]. Requirements for micronutrients increase during periconception and during pregnancy, while requirements for energy and protein increase particularly during the second and third trimesters [2,3].

Protein requirements during pregnancy were derived by the Institute of Medicine (IOM) using a factorial method that accounts for protein deposition and the additional protein required to support healthy weight gain during pregnancy [3]. In the first trimester of pregnancy, the additional protein needs are minimal at approximately 1 g/day and the estimated average requirement (EAR) is not different from the value of 0.66 g/kg bw/day for non-pregnant females [3]. In the second and third trimesters of pregnancy, protein needs increase an average of 21 g/day, resulting in an EAR of 0.88 g/kg bw/day [3]. Assuming a standard body weight for women, these EARs are equivalent to a daily protein intake of 38 g in the first trimester and 59 g in the latter two trimesters. The recommended dietary allowance (RDA) for protein during the first trimester of pregnancy is estimated at 46 g/day (0.8 g/kg bw/day), and at 71 g/day (1.1 g/kg bw/day) during the second and third trimesters [3]. 

Protein requirements established outside North America using the factorial method are similar to recommendations established by the IOM. The European Food Safety Authority (EFSA) identified additional protein needs of 1, 9, and 28 g/day, respectively by trimester of pregnancy [4], while D-A-CH, the nutrition societies of Germany, Austria, and Switzerland, set nutrient requirements for protein at 0.8, 0.9, and 1.0 g/kg bw/day during the first, second and third trimesters, respectively [5]. The indicator amino acid oxidation (IAAO) method has been studied as an alternative method to estimate protein requirements, including requirements during pregnancy, and tends to identify higher protein requirements relative to the balance or factorial methods (+15–73%) [5,6,7].

Nationally representative nutrient intake estimates for pregnant women in the U.S. are limited, and the available assessments of intake adequacy have largely focused on micronutrient intakes [8]. Using the National Health and Nutrition Examination Surveys (NHANES) 2001–2014, mean usual intake of protein by pregnant, non-lactating women in the U.S. was estimated at 81.9 g/day, though prevalence of protein intake below the EAR was not examined [8]. Based on data from 2007–2010, mean usual intake of protein by pregnant women in the U.S. was estimated at 78.6 g/day [9]; distributions of usual intake of protein among the sample indicated that the EAR for women in the second and third trimester of pregnancy fell between the 5th and 10th percentiles of intake, suggesting greater inadequacy of protein intake among women during pregnancy relative to inadequacy among non-pregnant women [10,11]. 

The purpose of this study was to assess protein intake and adequacy of intake among a representative sample of pregnant women in the U.S. population. To provide further insight on protein intake, sources of protein (i.e., plant and animal, including type of animal source) consumed by pregnant women were characterized.

## 2. Materials and Methods

### 2.1. Data Source and Study Population

This cross-sectional study was conducted with data collected in NHANES and the dietary recall component known as What We Eat in America (WWEIA) [12]. The NHANES provides nationally representative nutrition and health data used to develop prevalence estimates for nutrition and health status measures in the U.S. and serves as the foundation of many nutrient assessments for national nutrition policy [13]. Each NHANES cycle includes an in-person household interview, a health examination in a mobile examination center (MEC), and a telephone follow-up interview. Approval for the NHANES data collection was provided by the National Center for Health Statistics (NCHS) Research Ethics Review Board.

Pregnancy status was assessed with a urine pregnancy test during the MEC examination and women self-identified as pregnant reported month of pregnancy. More recent NHANES cycles have not included the question on month of pregnancy, thus this analysis was limited to survey cycles 2003–2004, 2005–2006, 2007–2008, 2009–2010, and 2011–2012. The sample population for this study was pregnant women ages 20–44 years as identified by a positive urine pregnancy test (*n* = 715), were not lactating (*n* = 704), self-identified month of pregnancy (*n* = 604), and provided two valid dietary recalls as determined by NCHS. The final sample of pregnant, non-lactating women included in this analysis is 528.

### 2.2. Population Characteristics

Demographic, lifestyle, and reproductive health characteristics were included to characterize the sample population by trimester of pregnancy and for completeness, women missing data on trimester of pregnancy. Self-reported month of pregnancy was categorized as trimester 1 (months 1, 2, 3), trimester 2 (months 4, 5, 6), and trimester 3 (months ≥ 7). Demographic characteristics include age at screening, race/ethnicity (non-Hispanic white, non-Hispanic black, Mexican American or other Hispanic, or other race), poverty income ratio (PIR; <1.85, ≥1.85), education status (less than high school, high school diploma, some college, undergraduate degree or higher), and marital status (married, widowed or divorced or separated, never married). Lifestyle characteristics include self-reported smoking status (never smoked, past smoker, current smoker), physical activity (<10 min/week, 10 to 150 min/week, ≥150 min/week), and self-reported dietary supplement use (Yes, No). Reproductive health characteristics include parity, defined as the number of live births, and gestational weight gain (GWG) classified as inadequate, adequate, or excessive using criteria established by the IOM [14]. Pre-pregnancy BMI was calculated from self-reported pre-pregnancy body weight collected during the in-home interview and standing height measured in the MEC. Usual intake of energy by trimester of pregnancy was also estimated.

### 2.3. Dietary Intake Data

The WWEIA component of NHANES consists of two 24-h dietary recalls (midnight to midnight); the first dietary recall is collected during the MEC interview and the second is collected during the follow-up telephone interview 3–10 days after the MEC interview. Each recall is collected by trained interviewers using USDA’s Automated Multiple-Pass method. For each food reported in WWEIA, the survey specific USDA Food and Nutrient Database for Dietary Studies (FNDDS) databases provide the translation of foods as reported consumed into gram amounts and ingredients for the processing of energy and nutrient intakes [15].

### 2.4. Assessment of Protein Intake and Adequacy

Estimates of protein intake were generated as grams per day (g/day) and as grams per kilogram pre-pregnancy body weight per day (g/kg bw/day) assuming actual body weight when it was within a normal healthy weight range (body mass intake (BMI) ≥18.5 and <24.9 kg/m^2^), or the pre-pregnancy body weight that would place the woman at the nearest endpoint of the healthy range. Adjustment of body weight to a value aligning with a healthy body weight is consistent with the approach used in assessments of protein intake and adequacy of intake [16,17]. Protein intake adequacy was defined as the prevalence of usual intake of protein below the EAR. 

Mean protein intake was also estimated as g/day and g/kg bw/day among pregnant women categorized by age, race/ethnicity, PIR, and education status. Given the small sample size in some subpopulations, this assessment was limited to demographic characterizations without further classification by trimester of pregnancy.

### 2.5. Identification of Protein by Source

The USDA food codes representing each food consumed were disaggregated into the survey-specific Standard Reference (SR) items used by USDA to process nutrient values [18]. Each SR code was then categorized as plant or animal, with animal sources further categorized as red meat (i.e., beef, pork, other), poultry, cured meat and poultry, seafood, eggs, and dairy. Contributions of protein by source were calculated per 100 g food code. For SR codes representing mixtures of protein sources, contributions from different sources were estimated from USDA data by translating each SR code into food pattern equivalents [19]. Protein intake by source was summed across all foods consumed over the two days of recall per respondent and divided by the 2-day average protein intake per person to develop protein intake (g/day) by source per person.

### 2.6. Assessment of Protein Adequacy Using Alternate Requirements

As an exploratory analysis, adequacy of protein intake among pregnant women in the U.S. was also assessed using alternate reference values for adequate protein intake during pregnancy developed from the IAAO method. These alternate EAR values correspond to intake of 1.22 g/kg bw/day during the early stages (~16 weeks) of pregnancy and 1.52 g/kg bw/day during late stages (~36 weeks) [6,7]. In this analysis, the alternate EAR during the first trimester of pregnancy was assumed to be 0.93 g/kg bw/day, which is the IAAO value for non-pregnant adults. The alternate EAR during the second trimester of pregnancy was assumed to be 1.22 g/kg bw/day, and the alternate EAR during the third trimester was assumed to be 1.52 g/kg bw/day. Consistent with the assessments of intake relative to the IOM EARs, the pre-pregnancy body weight for each pregnant woman in the sample was adjusted to a body weight that would place the woman at the nearest endpoint of the healthy range (≥18.5 to <24.9 kg/m^2^).

### 2.7. Statistical Analyses

Population characteristics of the sample of pregnant women by trimester of pregnancy were summarized and compared using ANOVA with Bonferroni adjusted *p*-values for multiple comparisons for continuous variables, and the Pearson Chi-square test for the categorical variables.

Usual intakes of protein and energy were estimated using the National Cancer Institute (NCI) method [20] and the SAS macros developed by NCI for modeling a single dietary component with covariates for day of the week (weekday/weekend) and sequence of the dietary recall (day 1 or 2), and usual intake of protein was also adjusted for age, race/ethnicity, and education status. Balanced repeated replicate weights (Fay adjustment factor = 0.3) based on 2-day dietary recall statistical weights were used for SE estimation [21]. Additionally, mean protein intake was estimated for subpopulations by demographic characteristics of age, race/ethnicity, PIR, and education status based on 2-day average intakes. 

Prevalence of usual intake of protein below the IOM EAR [3] and the alternate IAAO EAR reference values for protein adequacy [6,7] were calculated using the cut-point method [22]. Using the 2-day average protein intakes, mean percent contributions of protein intake from plant and animal sources were estimated for all pregnant women and by subcategory of trimester of pregnancy using the “population ratio” method [23]. Percent contributions from animal and plant sources were also calculated among pregnant women categorized by demographic characteristics of age, race/ethnicity, PIR, and education status. 

Logistic regression models were used to test the hypothesis that percent of total protein consumed as animal protein is associated with an increased odds of meeting the trimester-specific RDA for protein and increased likelihood of adequate GWG. In the analysis of GWG, women were classified in two categories: Adequate and inadequate (including excess) GWG. The logistic regression models were initially adjusted for the following covariates: Age, trimester of pregnancy, parity, race/ethnicity, marital status, education, family PIR, energy intake, pre-pregnancy BMI, smoking status, and physical activity. Non-statistically significant covariates were sequentially dropped from the models. Protein intake was represented by the 2-day average intake in the regression analyses.

Statistical analyses were completed using SAS (version 9.4, SAS Institute Inc., Cary, NC, USA) and STATA V12.1 (StataCorp, College Station, TX 77845, USA). All analyses used the appropriate statistical weights provided in NHANES to account for oversampling, survey non-response, and post-stratification and estimates of the standard errors (SEs) and confidence intervals were design adjusted. Estimates are presented as mean ± SE. Statistical significance was defined as Bonferroni-adjusted *p* < 0.05

## 3. Results

### 3.1. Population Characteristics

Of the 528 pregnant women included in this sample, 109 women were in trimester 1 of pregnancy, 207 were in trimester 2, and 212 were in trimester 3; 75 additional women were pregnant but were missing information on month of pregnancy (Table 1). With the exception of parity and GWG, population characteristics did not vary by trimester of pregnancy among women with a reported month of pregnancy; the population was primarily non-Hispanic white, married women, a PIR ≥ 1.85, at least some college education, 1 or more previous deliveries, achieved at least 10 min of moderate physical activity per day, were non-smokers, and reported intake of dietary supplements in the past 30 days. Compared to the 528 pregnant women with a reported month of pregnancy, the 75 pregnant women without trimester of pregnancy differed by education status and were less likely to use dietary supplements (*p* < 0.05). Usual intake of energy among the 528 pregnant women was 2272 ± 66.0 kcal/day, with intake by trimester 1, 2, and 3 of pregnancy at 2275 ± 107.2, 2298 ± 79.7, and 2247 ± 96.3 kcal/day, respectively.

### 3.2. Intake of Protein and Adequacy of Intake

Usual intake of protein was 88 ± 4.3, 82 ± 3.1, and 82 ± 2.9 g/day among women in trimester 1, 2, and 3 of pregnancy, respectively (Table 2), or 1.30 ± 0.10, 1.35 ± 0.06, and 1.35 ± 0.05 g/kg bw/day based on pre-pregnancy body weight in a healthy range. Similar results were observed for protein intake by trimester when adjusted for age, race/ethnicity, and education status (data not shown).

Adequacy of protein intake was assessed by calculating the percentage of women with usual intake of protein below the EAR. An estimated 4.5 ± 4.8% of women in the first trimester of pregnancy consumed less than the EAR of 0.66 g/kg bw/day; given the sample of 105 women in the first trimester of pregnancy, this estimate is potentially less statistically reliable [24,25]. Among women in the second and third trimesters of pregnancy, 12.1 ± 4.3% and 12.8 ± 4.6% of women, respectively, consumed less than the EAR of 0.88 g/kg bw/day.

Intake of protein by demographic factors of age, race/ethnicity, PIR, and education status did not differ among groups when assessed as 2-day average intake in g/day or g/kg bw/day based on pre-pregnancy body weight in a healthy range (Table 3). 

### 3.3. Intake of Protein by Source

Approximately two thirds (66.2 ± 0.9%) of total protein intake was from animal sources and ranged from 69.1 ± 1.3%, 64.9 ± 1.8%, and 65.3 ± 1.1% for women in the first, second, and third pregnancy trimester, respectively. Red meat, poultry, and cured meat and poultry combined accounted for approximately one-third of total intake of protein among pregnant women (34.7%) while dairy accounted for 23.1 ± 0.8% of protein intake. Protein from eggs and seafood each accounted for approximately 4–5% of protein consumed by pregnant women. Total red meat intake differed between groups (*p* < 0.05), though no consistent trend across the three pregnancy trimesters was observed in the contribution of other sources to total protein intake (Table 4).

Intake of percent protein as animal or plant protein among pregnant women categorized by demographic factors of age, race/ethnicity, PIR, and education status are summarized in Table 5. Statistically significant differences (*p*-value < 0.05) were observed in the percent of total protein consumed as animal or plant protein for race/ethnicity, PIR, and education status.

### 3.4. Animal Protein and Protein Requirements

After dropping the non-statistically significant covariates, the final logistic regression model evaluated the association between adequacy of protein intake relative to the RDA and percent protein from animal sources, and included covariates for pregnancy trimester, race/ethnicity, total energy intake, and pre-pregnancy BMI status. The odds ratio associated with percent protein from animal sources in the final model was 1.1 ± 0.02 (*p*-value < 0.05), indicating that pregnant women were more likely to meet the trimester-specific protein RDA as their percent protein intake from animal sources increased. Women with higher total energy intake and those in the first pregnancy trimester were also more likely to meet the protein RDA than those with lower energy intake or those in pregnancy trimester two or three. No statistically significant associations between adequate GWG and either meeting the protein RDA requirement or percent protein from animal sources were found in the final models (data not shown). 

### 3.5. Adequacy of Protein Intake Based on Alternate Requirements 

Adequacy of protein intake, based on pre-pregnancy body weight in a healthy range, was also assessed using alternate protein requirements derived for pregnant women with the IAAO method [6,7]. Protein intake among pregnant women was below the alternate EAR for 18.5 ± 10.0%, 40.1 ± 5.5%, and 66.9 ± 5.1% of pregnant women in trimester 1, 2, and 3, respectively (Figure 1).

## 4. Discussion

In this study, protein intake and adequacy of protein intake were examined in a nationally representative sample of pregnant women by trimester of pregnancy. The prevalence of inadequate protein intake among women in the first trimester of pregnancy was approximately 5% based on the EAR of 0.66 g/kg bw/day. Among women in the second or third trimester of pregnancy, 12–13% of women consumed inadequate levels of protein based on the EAR of 0.88 g/kg bw/day. Results from this study indicate that many pregnant women, in particular those beyond the first trimester of pregnancy, consumed suboptimal levels of protein relative to recommendations established by the IOM. 

The estimates of protein intake and adequacy of intake developed in this study are based on data collected in 2003–2012. While more recent NHANES data are available, data collected in recent NHANES cycles do not include information on trimester of pregnancy, consequently it would not be possible to compare intakes to trimester-specific requirements. A recent review of protein intake from USDA shows that total protein intake among U.S. females ages 20–39 years remained relatively unchanged between 2005–2006 (74 g/day) and 2015–2016 (73 g/day), and that the estimated contribution of animal sources to total protein intake was approximately 67% [26], which is consistent with the estimated contributions calculated in the current study. The stable intake of protein over the last decade and the consistent contribution of protein from animal sources suggest that the estimates of intake developed in this study may provide relevance for current protein intake and adequacy among pregnant women. The data also provide a point of reference for comparison in future analyses. 

The 2020–2025 Dietary Guidelines for Americans (DGA) are the first DGA to include nutrition recommendations for women during the life stages of pregnancy and lactation [27]. The Dietary Guidelines Advisory Committee (DGAC) did not assess adequacy of protein intake in units of g/kg bw/day, though intake of protein foods relative to dietary recommendations was examined [27,28]. Based on intakes in 2013–2016, the DGAC reported that nearly half (47%) of pregnant women consumed fewer ounce equivalents of protein foods than recommended, with particularly low intakes of seafood and plant-based protein [28]. However, the DGA did not identify protein as a nutrient of concern for pregnant women. Results in the current study show that percent protein intake from animal sources was positively associated with likelihood of meeting the trimester-specific RDA. 

The assessment of protein adequacy based on alternate requirements developed with the IAAO method indicate that approximately 40–67% of pregnant women consumed inadequate levels of protein during periods of significant fetal and maternal protein deposition. Adequacy of protein intake based on requirements derived with the IAAO method therefore suggest that suboptimal protein intake among pregnant women may be of greater concern than indicated by assessment with the IOM EAR. The IAAO method used to estimate protein requirements in pregnancy as reported by Stephens and colleagues is based on a 1-13C–labeled amino acid tracer in a short period of dietary intervention [6,7]. There has been debate about use of the IAAO method to determine protein requirements [29,30,31]. We included a comparison of protein intake to these requirements as an exploratory analysis to provide insight on the potential range of protein inadequacy among pregnant women. Further research on protein needs during pregnancy is needed.

Strengths of this study include a relatively large, nationally representative sample of pregnant women with identification by trimester of pregnancy that permitted comparisons of protein intake to trimester-specific requirements. Protein intake and adequacy of intake were based on usual intake estimates derived using established statistical methods and adjustment of body weight to a value falling within the range of a healthy BMI, which is consistent with assessment of protein adequacy [16,17]. Use of body weights without this adjustment would overestimate the percentage of women with inadequate intake of protein. Percent contributions of protein by source were based on 2-day average intakes, which may provide a more representative profile of typical intake than a single day of recall.

As previously noted, the data used in this analysis are not current, though evidence among non-pregnant women suggests that protein intake has not changed over the last decade. Other limitations of the analysis must be acknowledged. Estimates of protein intake on a body weight basis were calculated using self-reported pre-pregnancy weights, as measured pre-pregnancy body weight values are not available in NHANES due to the cross-sectional nature of the survey. Pregnant women missing the trimester of pregnancy were excluded from the sample population which could introduce bias. Due to sample size considerations, assessment of protein intake and intake from animal or plant sources across key demographic characteristics was based on the total sample of pregnant women and therefore may not reflect potential differences by trimester of pregnancy. Additionally, dietary recalls were self-reported and therefore subject to misreporting.

Findings from this study indicate that approximately one in eight women in the second and third trimesters of pregnancy have protein intakes below the EAR. Guidance for pregnant women should include advice to consume adequate levels of protein from a variety of protein sources. For women following a Healthy U.S.-Style Dietary Pattern, protein foods to encourage include lean meats, poultry, eggs, seafood, and plant-based proteins.

## Figures and Tables

**Figure 1 nutrients-13-00795-f001:**
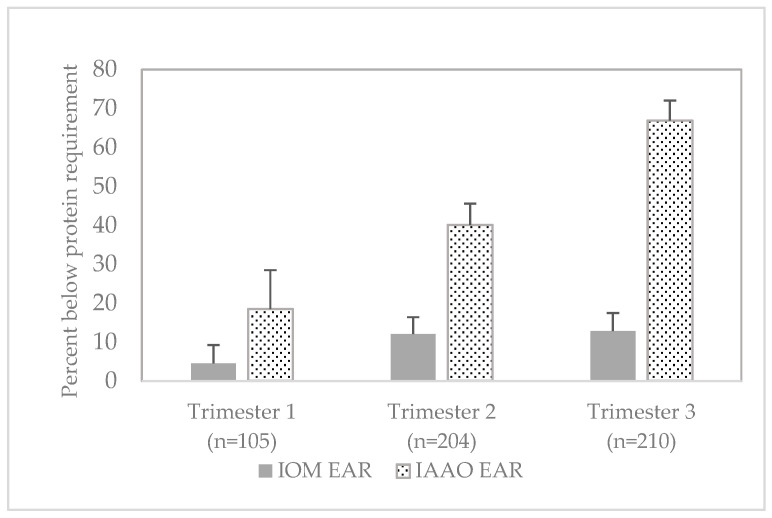
Percent of pregnant women ages 20–44 years with protein intake below reference requirements, 2003–2012 National Health and Nutrition Examination Survey. Values presented as mean percent (±SE). Protein requirements-IOM EAR: 0.66 g/kg bw/day for trimester 1, 0.88 g/kg bw/day for trimesters 2 and 3 [3]; IAAO EAR: 0.93 g/kg bw/day for trimester 1, 1.22 g/kg bw/day for trimester 2, 1.52 g/kg bw/day for trimesters 3 [6,7]. Abbreviations: EAR—estimated average requirement, IAAO—indicator amino acid oxidation, IOM—Institute of Medicine.

**Table 1 nutrients-13-00795-t001:** Characteristics of sample of pregnant women ages 20–44 years in 2003–2012 National Health and Nutrition Examination Survey.

Characteristic	Total with Trimester of Pregnancy ^1^ (*n* = 528)	Trimester 1 (*n* = 109)	Trimester 2 (*n* = 207)	Trimester 3 (*n* = 212)	Missing Trimester ^2^ (*n* = 75)
Age (years)	28.7 ± 0.44	27.9 ± 0.61	28.3 ± 0.69	29.6 ± 0.74	28.3 ± 1.07
Age (%)					
20–30 years	62.4 ± 3.76	75.0 ± 6.34	62.7 ± 5.64	53.6 ± 6.8	66.9 ± 7.53
31–44 years	37.6 ± 3.76	25.0 ± 6.34	37.3 ± 5.64	46.4 ± 6.8	33.1 ± 7.53
Race/ethnicity (%)					
Mexican American/ Other Hispanic	23.1 ± 3.26	24.2 ± 7.06	22.2 ± 4.76	23.2 ± 3.91	18.0 ± 6.08
Non-Hispanic white	52.9 ± 4.25	58.6 ± 8.86	47.3 ± 5.87	53.7 ± 5.84	46.3 ± 9.38
Non-Hispanic black	13.5 ± 2.68	8.7 ± 3.69	20.9 ± 4.41	10.6 ± 3.15	29.6 ± 7.46
Other race	10.5 ± 2.52	8.6 ± 4.13	9.7 ± 3.92	12.5 ± 4.15	6.1 ± 3.79
Poverty income ratio (PIR, %)					
<1.85	34.6 ± 3.92	32.6 ± 8.73	38.3 ± 5.49	32.8 ± 5.36	39.5 ± 8.58
≥1.85	65.4 ± 3.92	67.4 ± 8.73	61.7 ± 5.49	67.2 ± 5.36	60.5 ± 8.58
Education status (%)					
<High school	19.9 ± 2.47	14.1 ± 3.94	23.4 ± 5.28	20.8 ± 4.70	19.5 ± 6.83
High school diploma	13.7 ± 1.93	9.7 ± 3.19	15.1 ± 3.68	15.4 ± 3.28	34.9 ± 9.02
Some college	33.2 ± 3.34	47.7 ± 8.15	29.7 ± 4.69	26.3 ± 5.25	21.5 ± 5.8
Undergraduate degree or higher	33.2 ± 3.36	28.5 ± 7.18	31.9 ± 5.55	37.5 ± 6.39	24.0 ± 7.6
Parity (%)					
0	25.6 ± 3.55	31.7 ± 7.94	30.0 ± 5.53	18.3 ± 2.97	38.4 ± 12.74
1	43.5 ± 3.82	40.4 ± 7.10	51.8 ± 6.67	38.8 ± 6.10	25.5 ± 11.71
≥2	30.9 ± 3.20	27.9 ± 5.78	18.2 ± 3.24	42.9 ± 5.62	36.1 ± 10.85
Marital status (%)					
Married	69.3 ± 3.06	68.6 ± 6.74	68.9 ± 5.29	70.2 ± 5.00	49.1 ± 9.13
Widowed/divorced/separated	2.3 ± 0.69	3.7 ± 1.84	1.0 ± 0.43	2.3 ± 1.25	3.1 ± 1.39
Never married	28.4 ± 3.1	27.7 ± 6.62	30.1 ± 5.12	27.5 ± 4.98	47.8 ± 8.84
Physical activity, minutes/week (%)					
<10	31.7 ± 3.44	24.9 ± 6.41	34.4 ± 5.82	34.1 ± 5.86	51.2 ± 8.93
10 to <150	30.1 ± 3.57	35.2 ± 6.83	29.1 ± 4.97	27.5 ± 5.38	19.7 ± 4.51
≥150	38.2 ± 3.94	39.9 ± 7.26	36.5 ± 5.76	38.4 ± 6.74	29.1 ± 9.17
Smoking status (%)					
Never smoked	69.3 ± 3.53	69.5 ± 6.73	65.5 ± 6.33	72.4 ± 5.16	69.4 ± 8.92
Past smoker	23.3 ± 3.46	21.3 ± 6.15	26.7 ± 5.94	21.7 ± 4.72	17.4 ± 8.27
Current smoker	7.4 ± 1.48	9.2 ± 3.04	7.7 ± 3.25	6.0 ± 2.67	13.2 ± 5.87
Dietary supplement use (%)	85.6 ± 2.11	75.7 ± 5.76	88.6 ± 3.35	89.7 ± 3.79	43.8 ± 8.29
Pre-pregnancy body weight (kg)	70.7 ± 1.37	74.0 ± 3.03	72.6 ± 2.76	68.0 ± 1.97	-
Pre-pregnancy BMI (kg/m^2^)	26.8 ± 0.5	27.6 ± 1.0	27.7 ± 0.93	25.6 ± 0.79	-
Pre-pregnancy BMI status (%)					
Underweight	5.2 ± 1.7	4.1 ± 3.45	5.8 ± 2.41	5.4 ± 3.27	-
Normal weight	46.7 ± 3.57	40.6 ± 8.21	39.3 ± 5.31	57.0 ± 5.68	-
Overweight	21.8 ± 2.91	16.3 ± 6.07	25.7 ± 5.67	22.0 ± 4.16	-
Obese	26.3 ± 3.35	39.0 ± 8.11	29.1 ± 4.86	15.7 ± 3.56	-
GWG status (%)					
Inadequate	19.7 ± 2.87	35.8 ± 8.47	16.0 ± 3.78	12.2 ± 3.63	-
Adequate	31.5 ± 3.87	30.8 ± 8.45	30.4 ± 5.86	32.8 ± 6.62	-
Excess	48.9 ± 3.75	33.3 ± 7.47	53.6 ± 6.24	55.1 ± 6.73	-

Values reported as mean ± SE. ^1^ Pregnant women with self-reported month of pregnancy, excluding women missing data for PIR (*n* = 27), parity (*n* = 19), physical activity (*n* = 2), pre-pregnancy BMI (*n* = 9), and GWG (*n* = 9). ^2^ Pregnant women missing self-reported month of pregnancy, excluding women missing data for PIR (*n* = 4) and parity (*n* = 36). Pre-pregnancy body weight was not queried of these women. Statistically significant differences: Across trimester of pregnancy (trimester 1, 2, 3) for parity and GWG category (*p* < 0.05); among women with vs. without trimester of pregnancy for education status and dietary supplement use (*p* < 0.05).

**Table 2 nutrients-13-00795-t002:** Usual protein intake among pregnant women ages 20–44 years in 2003–2012 National Health and Nutrition Examination Survey.

Trimester of Pregnancy		Percentile						
	Mean	5th	10th	25th	50th	75th	90th	95th
	grams protein per day (g/day)							
Total (*n* = 528)	84 ± 2.3	57 ± 3.8	62 ± 3.3	71 ± 2.7	83 ± 2.3	95 ± 2.9	107 ± 4.2	115 ± 5.2
1 (*n* = 109)	88 ± 4.3	61 ± 5.4 ^1^	66 ± 5.2 ^1^	75 ± 4.8	87 ± 4.4	100 ± 4.7	112 ± 5.5^1^	120 ± 6.3 ^1^
2 (*n* = 207)	82 ± 3.1	55 ± 4.0	60 ± 3.6	70 ± 3.2	80 ± 3.0	92 ± 3.8	104 ± 5.1	111 ± 6.0
3 (*n* = 212)	82 ± 2.9	56 ± 4.1	61 ± 3.7	70 ± 3.2	82 ± 2.9	94 ± 3.5	105 ± 4.9	112 ± 5.9
	grams protein per kilogram body weight (g/kg bw/day) ^2^							
Total (*n* = 519)	1.34 ± 0.04	0.71 ± 0.10	0.82 ± 0.09	1.04 ± 0.07	1.31 ± 0.04	1.61 ± 0.05	1.88 ± 0.08	2.06 ± 0.10
1 (*n* = 105)	1.30 ± 0.10	0.68 ± 0.15 ^1^	0.79 ± 0.14 ^1^	1.00 ± 0.13	1.27 ± 0.11	1.57 ± 0.09	1.84 ± 0.09 ^1^	2.03 ± 0.09 ^1^
2 (*n* = 204)	1.35 ± 0.06	0.71 ± 0.09	0.84 ± 0.08	1.06 ± 0.06	1.32 ± 0.05	1.61 ± 0.08	1.90 ± 0.11	2.07 ± 0.13
3 (*n* = 210)	1.35 ± 0.05	0.72 ± 0.09	0.83 ± 0.08	1.05 ± 0.06	1.33 ± 0.05	1.62 ± 0.07	1.89 ± 0.11	2.07 ± 0.13

Values reported as mean ± SE. ^1^ Values may be less statistically reliable based on guidance from NCHS [24] and reported variance inflation factors up to 2.51 in NHANES 2005–2012 [25]. ^2^ Intake of protein as g/kg bw/day calculated from pre-pregnancy body weight assuming reported body weight when it was within a normal healthy weight range (BMI ≥18.5 and <24.9), or the weight that would place the woman at the nearest endpoint of the healthy range.

**Table 3 nutrients-13-00795-t003:** Protein intake by demographic characteristics among pregnant women ages 20–44 years in 2003–2012 National Health and Nutrition Examination Survey.

		Protein Intake	
Characteristic	*n* ^1^	g/day	g/kg bw/day ^2^
Total with Trimester of Pregnancy	528/519	84 ± 1.9	1.31 ± 03
Age (years)			
20–30	356/348	86 ± 2.3	1.41 ± 0.04
31–44	172/171	82 ± 2.7	1.32 ± 0.04
Race/ethnicity			
Mexican American/Other Hispanic	180/175	87 ± 3.4	1.45 ± 0.06
Non-Hispanic white	239/237	81 ± 3.0	1.28 ± 0.05
Non-Hispanic black	72/70	86 ± 4.0	1.41 ± 0.08
Poverty income ratio (PIR)			
<1.85	222/218	86 ± 2.8	1.42 ± 0.04
≥1.85	279/275	83 ± 2.6	1.35 ± 0.04
Education status			
<High school	136/135	82 ± 3.5	1.37 ± 0.06
High school diploma	96/92	88 ± 5.2	1.51 ± 0.11
Some college	156/153	83 ± 3.6	1.32 ± 0.07
Undergraduate degree or higher	140/139	86 ± 3.3	1.38 ± 0.05

Values reported as mean ± SE; 2-day average intakes. ^1^ Sample size for protein intake g/day and g/kg bw/day, respectively. ^2^ Intake of protein as g/kg bw/day calculated from pre-pregnancy body weight assuming reported body weight when it was within a normal healthy weight range (BMI ≥18.5 and <24.9), or the weight that would place the woman at the nearest endpoint of the healthy range. No statistically significant differences among groups were observed.

**Table 4 nutrients-13-00795-t004:** Mean percent of protein intake by source among pregnant women ages 20–44 years in 2003–2012 National Health and Nutrition Examination Survey.

Protein Source		Total (*n* = 528)	Trimester 1 (*n* = 109)	Trimester 2 (*n* = 207)	Trimester 3 (*n* = 212)
		% of Protein			
Animal	Total	66.2 ± 0.9	69.1 ± 1.3	64.9 ± 1.8	65.3 ± 1.1
	Total red meat ^1^	15.5 ± 1.2	18.2 ± 2.7	11.7 ± 1.3	16.6 ± 1.9
	Beef	12.7 ± 1.1	14.8 ± 2.7	10 ± 1.1	13.3 ± 1.7
	Pork	2.5 ± 0.5	2.1 ± 0.8	1.7 ± 0.5	3.3 ± 0.9
	Poultry	13.2 ± 1.1	12.6 ± 2.2	13.9 ± 1.6	13.1 ± 1.8
	Cured meat and poultry	6.0 ± 0.5	5.7 ± 0.9	6.5 ± 1.0	5.8 ± 1.0
	Dairy	23.1 ± 0.8	22.7 ± 1.6	25.2 ± 1.5	21.7 ± 1.1
	Egg	4.2 ± 0.3	4.9 ± 0.9	3.7 ± 0.4	4.0 ± 0.5
	Seafood	4.3 ± 0.7	5.0 ± 1.9	3.8 ± 0.9	4.1 ± 1.0
Plant	Total	33.8 ± 0.9	30.9 ± 1.3	35.1 ± 1.8	34.7 ± 1.1

Values reported as mean percent ± SE. ^1^ Total red meat includes beef, pork, and other red meat (intake of other red meat was reported by three women). Total red meat intake differed among groups (*p* < 0.05).

**Table 5 nutrients-13-00795-t005:** Protein intake from animal vs. plant source by demographic characteristics among pregnant women ages 20–44 years in 2003–2012 National Health and Nutrition Examination Survey.

Characteristic	*n*	% of Protein by Source	
		Animal	Plant
Age (years)			
20–30	356	67.4 ± 1.0	32.6 ± 1.0
31–44	172	64.2 ± 1.4	35.8 ± 1.4
Race/ethnicity			
Mexican American/Other Hispanic	180	67.9 ± 1.0	32.1 ± 1.0
Non-Hispanic white	239	65.3 ± 1.4	34.7 ± 1.4
Non-Hispanic black	72	70.6 ± 1.2	29.4 ± 1.2
Poverty income ratio (PIR)			
<1.85	222	69.5 ± 1.5	30.5 ± 1.5
≥1.85	279	64.6 ± 1.2	35.4 ± 1.2
Education status			
<High school	136	68.7 ± 1.2	31.3 ± 1.2
High school diploma	96	70.8 ± 1.4	29.2 ± 1.4
Some college	156	67.1 ± 1.7	32.9 ± 1.7
Undergraduate degree or higher	140	62.1 ± 1.5	37.9 ± 1.5

Statistically significant differences (*p*-value < 0.05) were observed for race/ethnicity, PIR, and education status.

## Data Availability

The NHANES data described in the article and used in the analysis are publicly available from the CDC and USDA via: https://wwwn.cdc.gov/nchs/nhanes/ContinuousNhanes/Default.aspx (accessed on 1 March 2020) and https://www.ars.usda.gov/northeast-area/beltsville-md-bhnrc/beltsville-human-nutrition-research-center/food-surveys-research-group/, respectively (accessed on 1 March 2020).

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
