# Peer review of "Adequacy and Sources of Protein Intake among Pregnant Women in the United States, NHANES 2003–2012"

_nutrients, 2021, doi:10.3390/nu13030795_

Round 1
Reviewer 1 Report
Murphy and colleagues reported protein consumption according to trimesters in pregnant women from the NHANES study. Notably, more than 12% of pregnant women in the second and third trimesters were deficient in protein consumption. The results could provide precious fundamental information. This manuscript is well described; however, several aspects should be revised.
Major point
- The information on pre-pregnancy body weight is crucial in the present study; can the authors provide the average body weight and average BMI as well as the percentage of the BMI categories?
- Tables 2, 3, and Figure 1 are essential statistical informative data. Do these results alter by race, age, or other characteristics? Specifically, can the authors provide the results for subgroups or the results adjusted for age, race, and other potential confounders? Identifying the populations at higher risk of protein consumption deficiency is considered to be important.
- The authors should provide a note in the discussion section to help readers understand the differences in results between the IOM and IAAO methods.
- Could the authors provide the data of pregnant women excluded due to the trimesters' information deficit?
Minor point
- To clarify the sample size, please add the number of subjects to Figure 1.
Reviewer 2 Report
Authors using data from the National Health and Nutrition Examination Surveys examined protein intake and sources in pregnant women by trimester of pregnancy. This is a very well written manuscript with results clearly presented in tables. While below are a few methodological concerns, these are mainly details that probably have little impact on the results. This manuscript provides important information regarding protein intake in those in the latter two trimesters of pregnancy, namely over 10% of this population may not be getting the recommended level of protein.
Lines 104-108
Please justify only using subjects having “two valid dietary recalls.” The NCI method allows analyses of individuals with only a single day of intake as long as an adequate sample of subjects have two days of intake are available to estimate variance components to simulate distributions of intake. The number of subjects eliminated due to this decision (i.e., number of subjects with a single dietary recall) should be reported.
Lines 127-130
Authors are lauded for their effort to obtain a pre-pregnancy weight and thus BMI for subjects. Additionally, their approach to adjust weight to more ideal weights is also appropriate. This reviewer suggests if authors have analyses where they used actual pre-pregnancy body weight this would make for an excellent comparison of what happens to % of population below recommended protein intakes when actual body weights are used (predict percentage below the EAR will be considerably higher)
Lines 163-166
Authors report approach to assess percentage contribution of types of protein and appear to be determining percentages per person and then analyzing these values for a population percentage. The term used for this approach is “mean of ratios.” Most experts agree the better approach is to determine the “population ratio” which is the sum of source across the population divided by the total consumption of item of interest. This can be accomplished using PROC RATIO in several of the standard statistical programs used for NAHNES analyses. Suggest authors consider using this preferred approach (or at least add to the limitation section regarding this approach compared to the recommended approach). For more information see
Freedman, LS; Guenther, PM; Krebs-Smith, SM; Kott, PS (2008). A population’s mean Healthy Eating Index-2005 scores are best estimated by the score of the population ratio when one 24-hour dietary recall is available. J Nutr 138:1725-29
Lines 193-195
This reviewer was surprised 2-day weights were used as typically the one-day weights are used as the second day of intake is only used to estimate variance components needed to estimate distribution of intake. Authors should justify this use and/or identify if this approach yielded very different results (unlikely, but…).
Lines 383-385
Authors indicate the food groups typically recommended to increase protein intake. Given the expertise of this group, this reviewer is wondering if an analyses was conducted looking specifically at intakes of the FPED (Food Patterns Equivalent Databse) protein components. These results would be excellent to include as more specific recommendations on which food groups pregnant women need to increase could have been presented (if these data are not currently available maybe this can be considered in a future study).
Round 2
Reviewer 1 Report
I commend the authors for the appropriate corrections. Notably, the revised tables and figure provide the reader with very useful information.